evolution/biomaterials/ecology

hybridization, iridescence, signal evolution, ecological niche modelling

**Author for correspondence:**
Chad M. Eliason
e-mail: celiason@fieldmuseum.org

# Interspecific hybridization explains rapid gorget colour divergence in *Heliodoxa* hummingbirds (Aves: Trochilidae)

Chad M. Eliason[1,2], Jacob C. Cooper[1,4,5], Shannon J. Hackett[1,3], Erica Zahnle[5], Tatiana Z. Pequeño Saco[6], Joseph Dylan Maddox[3,7], Taylor Hains[1,3], Mark E. Hauber[8] and John M. Bates[1]

[1]Negaunee Integrative Research Center, [2]Grainger Bioinformatics Center, and [3]Pritzker Laboratory for Molecular Systematics and Evolution, Field Museum of Natural History, 1400 S Lake Shore Drive, Chicago, IL 60605, USA
[4]Committee on Evolutionary Biology, University of Chicago, 1025 E 57th Street, Chicago, IL 60637, USA
[5]Biodiversity Institute, University of Kansas, 1345 Jayhawk Boulevard, Lawrence, KS 66044, USA
[6]Directora de Monitoreo y Evaluacion de Recursos Naturales del Territorio, Plataforma digital única del Estado Peruano
[7]Laboratorio de Biotecnología y Bioenergética, Universidad Científica del Perú, Iquitos, Perú
[8]Department of Evolution, Ecology, and Behaviour, School at Integrative Biology, University of Illinois, Urbana-Champaign, IL 61801, USA

 CME, 0000-0002-8426-0373; JCC, 0000-0003-2182-3236; SJH, 0000-0002-1404-0332; MEH, 0000-0003-2014-4928; JMB, 0000-0002-5809-5941

Hybridization is a known source of morphological, functional and communicative signal novelty in many organisms. Although diverse mechanisms of established novel ornamentation have been identified in natural populations, we lack an understanding of hybridization effects across levels of biological scales and upon phylogenies. Hummingbirds display diverse structural colours resulting from coherent light scattering by feather nanostructures. Given the complex relationship between feather nanostructures and the colours they produce, intermediate coloration does not necessarily imply intermediate nanostructures. Here, we characterize nanostructural, ecological and genetic inputs in a distinctive *Heliodoxa* hummingbird from the foothills of eastern Peru. Genetically, this individual is closely allied with *Heliodoxa branickii* and *Heliodoxa gularis*, but it is not identical to either

when nuclear data are assessed. Elevated interspecific heterozygosity further suggests it is a hybrid backcross to *H. branickii*. Electron microscopy and spectrophotometry of this unique individual reveal key nanostructural differences underlying its distinct gorget colour, confirmed by optical modelling. Phylogenetic comparative analysis suggests that the observed gorget coloration divergence from both parentals to this individual would take 6.6–10 My to evolve at the current rate within a single hummingbird lineage. These results emphasize the mosaic nature of hybridization and suggest that hybridization may contribute to the structural colour diversity found across hummingbirds.

## 1. Introduction

Hybridization between species is a potential source of morphological [1], functional [2] and communication signal novelty [3,4]. Novel phenotypes can arise in hybrids whenever recombination is heterogeneous across the genome [5]. One particular form of segregation, transgressive segregation, in which recombination occurs in genes with antagonistic effects (e.g. the agouti-Mc1r system) [6], commonly produces distinctive, divergent phenotypes from both parentals [7]. While we have an understanding of the genetic and phenotypic effects of mutation and hybridization on the level of the individual [5,7], we know relatively little of how these factors might cause changes across different levels of biological scales and organization, including from tissue nanostructures through populations to species [8]. This information is critical for our understanding of how evolution acts on phenotypes, as different genetic mutations can result in the same observable phenotype [9]. Many of the most prominently studied traits in nature are sexually selected ornamental or communication traits [10]. Signal diversity often stems from variation in underlying morphological traits, such as dewlap muscle biomechanics [11] and melanosome morphology [12]. Signal traits are therefore a tractable system to study the effects of hybridization on character state evolution across biological scales.

Avian colours are produced by a combination of two general mechanisms: light absorption by pigments (pigment-based colours) and light scattering by organized feather nanostructures (structural colours) [13–15]. In the latter, morphological complexity of nanostructures suggests there may not be a clear one-to-one relationship between feather or integument morphology and colour phenotypes. For example, novel hybrid plumage coloration in *Lepidothrix* manakins results from an interaction between pigment-based (carotenoids) and structural colour mechanisms (spongy keratin-air nanostructures in feather barbs) [3]. In this case, hybrids are intermediate in most key physico-chemical traits, with the exception of the thickness of the pigment-laden outer cortex of feather barbs, suggesting that signal novelty in manakins is not a direct consequence of hybridization but rather is owed to protracted changes as pigments are gained later, after the evolution of a thickened barb cortex. Furthermore, colour diversity in the *Lepidothrix* system stems from combinatorial colour mechanisms (i.e. carotenoid pigments and nanostructures). To our knowledge, there are few examples documenting variation in coloration in closely related species or populations caused entirely by changes in feather nanostructure (i.e. structural colours).

Hummingbirds are known for their species diversity and diverse, vibrant colours [16,17]. This colour variability stems from changes in complex colour-producing nanostructures within feathers that can vary in at least six morphological parameters [17]. Both intersubspecific and interspecific hybridization is common in hummingbirds, with several studies describing the effects of hybridization on feather coloration [18–24]. At the same time, of the *ca* 180 species of hummingbirds with complex nanostructures leading to iridescent coloration in the throat (i.e. gorget), there are no known examples of intraspecific variation in gorget colour that are not tied to geography. These features make hummingbirds a productive study system in which to understand the phenotypic effects and evolutionary impacts of hybridization across levels of biological organization (e.g. feather nanostructure and colour signal phenotype).

In 2013, we conducted fieldwork in Parque Nacional de Cordillera Azul, San Martín, Peru and captured a male *Heliodoxa* hummingbird with novel colour patterns particularly with respect to its gorget. Here, we describe nanostructural colour analyses to characterize the colour patterns in the gorget. We use optical modelling to understand the physical mechanisms by which colour phenotypes arise. We combine this work with extensive phylogenetic analyses to understand the genetic origin of this individual, with specific emphasis on determining what role hybridization played. We also present ecological niche modelling to better understand a potentially important contact zone between

**Figure 1.** Gorget colour and feather nanostructural differences in *Heliodoxa* hummingbirds. Images show iridescent gorget coloration in the two pure species, *Heliodoxa gularis* (*a*) and *H. branickii* (*c*), as well as the putative hybrid individual FMNH 511084 (*b*). Reflectance spectra (middle panels) were taken at the angle of maximum reflectance (see electronic supplementary material, table S1 for list of specimens) and show a distinct lack of peak at the 450 nm wavelength for the hybrid. Lower panels are transmission electron microscopy (TEM) images of nanostructures responsible for iridescent colour production (scale bars are 500 nm). Note thinner cortex and thicker top platelets in the hybrid (see labels in (*e*)). Photo credits: Chad M. Eliason.

*Heliodoxa* taxa in Peru. Finally, we use character reconstruction and evolutionary rates to understand how novel phenotypes generated within (or between) species, relate to species-level morphological evolution in hummingbird gorget colour space. We argue that this example has implications for understanding the origins and rate of change of diversity in natural populations generally.

## 2. Methods

### 2.1. Specimen sampling

For two weeks in November 2013, a team headed by J.M.B. surveyed and captured birds in undisturbed forests of the foothills of Parque Nacional de Cordillera Azul, Peru. Mist-net lines were run up forested hillsides on both banks of the Río (River) Pescadero (a north flowing tributary of the larger Rio Huallaga which it enters from the east). On 23 November 2013, a male *Heliodoxa* hummingbird with a novel gorget colour (figure 1*b*) was captured and collected on the western bank of the Rio Pescadero at 953 m.a.s.l. (10.694° S, 13.422° W) and prepared as a skin with preserved tissue and catalogued as Field Museum of Natural History (FMNH) 511084. The gonads were recorded as $1 \times 1$ mm testes and moult was

noted on the body, wings and tail. The bird was photographed live in hand by J.M.B., who noted the throat iridescence seemed unusual (i.e. yellower) than that typical of *H. branickii*, the expected species in the region. In addition to the tissue samples used in genetic analyses, we also studied *H. branickii* (N = 11) and *H. gularis* (N = 5) adult male specimens from the FMNH collection and specimens on loan from the Louisiana State University Museum of Natural Sciences (LSUMNS) for spectral analysis and visual modelling (see electronic supplementary material, table S1 for specimen details). Three feathers from FMNH specimens, including of the atypical individual, were used for morphological analysis (see below).

## 2.2. Spectral analysis

We used an Ocean Optics USB 2000 UV-Vis spectrophotometer to measure reflectance spectra across bird-visible wavelengths of light (300–700 nm) relative to white and black standards. Since iridescent colours are, by definition, highly angle-dependent, we measured reflectance using a bifurcated fibreoptic probe with both the light source and spectrophotometer probe at the same 90° angle with respect to the feather (i.e. normal incidence), as well as at the angle for which we observed maximal reflectance, which was variable among specimens. The latter measurement geometry has been shown to be more reliable, especially for iridescent plumages [25], therefore we used spectra recorded at the optimal incidence angle for downstream analyses. From each spectrum, we determined hue (i.e. wavelength at peak reflectance) with the peakshape function in pavo [26] using the R statistical platform [27]. To compare perceptual distribution in avian colour space, we ran visual models assuming an ultraviolet-sensitive (UVS) hummingbird visual system [28] in pavo. We then used the bootcoldist function to assess the significance of divergence between populations of *H. gularis* and *H. branickii* in tetrahedral colour space versus the hybrid. Default values were used for cone ratios (1 : 2 : 2 : 4) and Weber fraction ($\omega = 0.1$).

## 2.3. Morphological analysis

To understand the morphological (nanostructural) traits underlying divergence in iridescent coloration, we used a transmission electron microscope (TEM; see below) to image cross-sections of three to five iridescent feather barbules from three body regions (crown, gorget and tail) per single individual of each parental species and the hybrid, following the protocol developed by Shawkey *et al.* [29]. Briefly, we dissected iridescent feather barbs, dehydrated them in ethanol, infiltrated the feathers with Embed 812 resin, and cured them at 60°F. We then sectioned the polymerized blocks into approximately 90 nm sections and imaged them on a Philips CM200 TEM at the Beckman Institute for Advanced Science and Technology at the University of Illinois at Urbana-Champaign. On each TEM image, we measured several traits known to be involved in iridescent colour production [17]: (i) melanosome platelet thickness (pt), (ii) amount of keratin between melanosomes (ker), (iii) air space diameter (air), (iv) number of melanosome layers, (v) thickness of top surficial melanosomes ($pt_{top}$), and (vi) keratin cortex thickness (cortex). For air space diameter, we took measurements both parallel ($air_{par}$) and perpendicular to the feather barbule surface ($air_{perp}$). This was done to determine whether deformation had occurred during sectioning (e.g. the resin often 'pulls away' from the outer barbule surface resulting in perpendicular deformation). We made the assumption that air spaces should be roughly isometric in cross-section, thus we took the average of perpendicular and parallel measurements. In total, we took 2421 individual measurements from 33 TEM images (9–13 images per taxon).

## 2.4. Optical modelling

Following previous work [17], we used a one-dimensional optical model to simulate reflectance. The model first creates vertical 'slices' of the feather nanostructure, all with a uniform refractive index, and then calculates light reflection as a function of wavelength at each interface using a transfer matrix approach [30]. Given that slight nanostructural variation within or among feather barbules can result in large colour differences [31], we modelled reflectance spectra using the average nanostructural parameters for each feather TEM image rather than the mean value for an individual. In previous work, we modelled reflectance spectra assuming spherical air spaces within melanosomes [17]. However, given the more rectangular shape observed in the *Heliodoxa* species analysed here (electronic supplementary material, figure S1), we also modelled reflectance spectra assuming block-shaped air spaces [32]. For all models, we used empirical values for the wavelength-dependent refractive indices of eumelanin [33] and keratin [34].

## 2.5. Quantifying phenotypic divergence

To quantify the amount and direction of phenotypic divergence between the putative hybrid and parental species, we used a recently developed method for calculating phenotypic divergence in F1 hybrids [35]. Briefly, we calculated the average distance from the midpoint of the line connecting the two parental phenotypes in two-dimensional space ($d_{\text{parent-bias}}$), with values close to unity suggesting similarity to one of the parental species and values of zero suggesting intermediate phenotypes. We also calculated the orthogonal divergence from the parental transect line ($d_{\text{mismatch}}$), with values near zero representing more intermediate phenotypes and values greater than zero indicating more transgressive phenotypes. To further compare the amount of colour divergence in the putative hybrid relative with the colour variability across hummingbirds in an evolutionary context, we used a published spectral dataset [17] and comprehensive hummingbird phylogeny [36] to estimate a multivariate rate of colour evolution using three-dimensional colour space coordinates as input. Given the significant level of phylogenetic signal in the colour data ($K = 0.53$, $p < 0.01$) estimated with physignal [37], we calculated scaled independent contrasts using the pic function [38] for each individual XYZ colour space coordinate, following McPeek *et al.* [39], and then calculated the average value of the squared contrasts at each node under a Brownian motion model. The average across all nodes of the tree yields an overall multivariate rate of evolution in units of $\Delta S^2 \text{ My}^{-1}$. Using this estimate, we determined the time needed to obtain the observed colour differences between the hybrid and parental species as the squared Euclidean distance in tetrahedral colour space ($\Delta S^2$) divided by the multivariate evolutionary rate.

## 2.6. Genomic sequencing

We extracted genomic DNA from FMNH 511084 using a Qiagen DNeasy Blood & Tissue Kit. We fragmented genomic DNA via sonication (Covaris M220), prepared the library following Glenn *et al.* [40], and enriched UCEs [41] using a MYbaits capture kit (Tetrapods 5 K v. 1, Arbor Biosciences) following the manufacturer's instructions. The UCE library was then sequenced on an Illumina MiSeq with a $2 \times 150$ Micro Kit v. 2.

## 2.7. Ultraconserved element variant calling and alignments

We combined genomic data from FMNH 511084 with ultraconserved element (UCE) data for members of the genus *Heliodoxa* [42]. We used the PHYLUCE pipeline [43] on raw reads to call SNPs and assemble UCE alignments using default parameters. For each UCE alignment, we estimated a maximum-likelihood tree using IQ-TREE 2 [44]. We then input these 3763 trees into splitsTree v. 4.18.3 [45] and constructed a phylogenetic network using ConsensusNetwork with an edge threshold of 0.1, following Caparros and Prat [46].

## 2.8. Estimating hybrid ancestry

To include off-target loci and estimate the hybrid ancestry of FMNH 511084 (i.e. the proportion of the genome with *H. branickii* or *H. gularis* ancestry), we mapped cleaned reads for *H. branickii*, *H. gularis* and the putative hybrid to the Anna's hummingbird (*Calypte anna*) genome [47]. We then called SNPs with bcftools consensus/call and filtered the dataset to only include SNPs with greater than 5× coverage and quality scores Q > 20. We further retained only fixed SNPs in each parental species (i.e. filtered out heterozygous sites in parentals) using the vcfR R package [48]. With this final dataset of 2131 SNPs, we used the R package introgress [49] to calculate hybrid index and interspecific heterozygosity and plotted these data using triangle.plot (see github for R code).

## 2.9. Ecological niche modelling

Occurrence records for *H. branickii* and *H. gularis* were downloaded from eBird [50] and GBIF (dois:10. 15468/dl.ufgoqv and 10.15468/dl.sctfy6), and concatenated into a single data file of unique localities using the R packages auk v. 0.4.3 [51] and tidyverse v. 1.2.1 [52]. Data were taken 'as is' with obvious spatial errors removed (namely, two records from the western slope of the Andes for *H. gularis*). Given spatial biases in the data due to accessibility, we opted for the use of presence-only based minimum volume ellipsoids (MVEs) [53] to generate estimates of species' niches. Environmental data were extracted using the R package raster v. 3.0–7 [54] from the ENVIREM [55] dataset for annual

**Figure 2.** Geographical distributions and niche divergence in *Heliodoxa* hummingbirds. *Heliodoxa gularis* and *H. branickii* are found along the eastern foothills of the Andes from Colombia to Bolivia, roughly separated by the Huallaga River of Peru: species distribution models derived from minimum volume ellipsoids of the species' ecological niches recreates this disjunct distribution well, with few areas north of the Huallaga suitable for *Heliodoxa branickii* and few areas south of the river suitable for *H. gularis* (*a*). The species diverge ecologically as well, with the hybrid individual being found both in an intermediate geographical locality (*a*) and intermediate environmental regime (*b*). Images modified from [81].

precipitation, precipitation seasonality and continentality. Elevation data were similarly extracted from the EarthEnv median GMTED2010 elevational dataset [56]. All variables were downloaded at a 30 arcsecond (*ca* 1 km at the equator) resolution. We used QGIS 3.10 (qgis.org) to create custom **M** dispersal regions for each species [57–59], with each **M** including the area of potential overlap in the Cordillera Azul region. A larger combined area was also used for performing niche equivalency tests. Minimum volume ellipsoids were defined using Mahalanobis distances and created using custom R scripts and the R package MASS [60]. Niche models were thresholded at 90% data inclusion to account for inaccurate plotting of data and the potential of some records to come from vagrant individuals [59]. Niche equivalency tests were performed using the methodology outlined by Warren *et al*. [61], wherein random occurrence points in each **M** were selected via custom R code by Cooper and Barragán [62] using the R package maptools [63] to create random pseudomodels. Each set of random pseudomodels was compared with the true model of the other species to create test distributions against which the comparison of the true MVE models could be compared. Comparisons were quantified using Schoener's *D* via the R package dismo v. 1.1–4 [64]. Furthermore, environmental data were analysed via principal components analyses (PCA) using the R package vegan v. 2.5–6 [65], and visualized in ggplot2 [66].

## 3. Results

Cordillera Azul National Park encompasses a large outlying highland that is separated to the east of the main Andes mountains by the large Río Huallaga which exists eastward into the Amazon Basin just beyond the northern end of the park, but which also defines the entire western edge of the park relative to the main Andean slopes. The rufous-webbed brilliant (*H. branickii*) reaches the northern limit of its distribution in the Cordillera Azul (it is not known this far north on the main eastern slope of the Andes), whereas the pink-throated brilliant (*H. gularis*) has a distribution that extends along the

**Table 1.** Quantifying transgressive feather phenotypes in hybrid hummingbirds. Values are average pairwise distances between all sets of traits. Parent-bias distance refers to the similarity of the hybrid to one of the two parental species (values of 0 indicate intermediate traits and values of 1 suggest the hybrid is exactly like one of the two parental species in multivariate space). Mismatch distance refers to the distance from the line connecting two parental phenotypes (0 falls on line, >1 more than distance between parental species in phenotypic space). See Thompson *et al.* [35] for methodological details.

| trait | patch | $d_{parent-bias}$ | $d_{mismatch}$ |
|---|---|---|---|
| morphology | gorget | 3.58 | 3.92 |
| | crown | 3.35 | 3.47 |
| | tail | 0.79 | 0.65 |
| empirical colour | gorget | 39.21 | 40.42 |
| | crown | 1.95 | 1.14 |
| | tail | 4.76 | 3.26 |
| modelled colour | gorget | 24.04 | 25.04 |
| | crown | 3.58 | 3.99 |
| | tail | 0.84 | 0.23 |

eastern slopes of the Andes with a southern limit at the Río Huallaga (figure 2*a*). The overall similarities of these two species had led to past speculation that they might hybridize in unsampled intervening areas [67], including the Cordillera Azul and the adjacent Andean slope to the west.

## 3.1. Colour divergence is greatest in the gorget

We used UV-Vis spectrophotometry and avian visual models to quantify coloration in males of each parental species and FMNH 511084. *Heliodoxa gularis* and *H. branickii* were significantly divergent in avian tetrahedral colour space for all measured plumage patches (just noticeable difference, JND > 1; electronic supplementary material, figures S2 and S3). Colour divergences were similar for crown, gorget and tail feathers (JNDs approx. 5; electronic supplementary material, figure S3). The greatest colour difference in FMNH 511084 was in gorget feathers, with crown and tail feathers being less divergent with respect to either parental species (electronic supplementary material, figure S3).

## 3.2. Morphological divergence in key colour-producing traits

To understand the morphological basis for the distinct gorget coloration of the hybrid, we used transmission electron microscopy (TEM) and image analysis [68]. For gorget feathers, FMNH 511084 had thicker melanosomes (both surficial and deeper into the feather barbule), more air within melanosomes and a thinner keratin cortex (tables 1 and 2). For crown feathers, the hybrid had solid (i.e. lacking air) surficial melanosomes and was intermediate in air spacing between the two parentals (figure 3, table 2). The cortex was thinner than both parental species. Tail feathers revealed that the hybrid was *H. branickii*-like in nearly all morphological traits (figure 3*f*, table 2). Thicker melanin layers in *H. gularis* (table 2) may explain its greener tail (as opposed to blue) coloration (figure 4*f*). These measurements were significantly repeatable for all traits considered (electronic supplementary material, figure S4).

## 3.3. Optical modelling supports empirical colour results

Given the known relationship between feather morphology and iridescent colour in hummingbirds [17], we hypothesized that divergence in morphology would be sufficient to explain the observed colour differences in the hybrid and parental species. Optical models based on morphological dimensions captured from TEM analysis mostly recapitulated the patterns observed in empirical spectra, with a few exceptions. Tail feathers in *H. gularis* were greener than *H. branickii* and the backcross hybrid (figure 4*c*). Similarly, crown feathers of *H. gularis* were greener and the hybrid was more blue-green (i.e. *H. branickii* like; figure 4*a*). The models predicted drabber crown feathers in *H. gularis* (but only for sphere-shaped air space models; electronic supplementary material, figure S5). The 'double-peak'

**Table 2.** Summary of feather nanostructure traits. Means (and 95% confidence intervals) are given in nm, with the exception of the number of layers. Note that confidence intervals represent within-individual variation, not within-species variation since these data are taken from single individuals. Traits for which the hybrid is transgressive are highlighted in italics.

| trait | patch | *Heliodoxa branickii* | hybrid | *H. gularis* |
|---|---|---|---|---|
| air space diameter | crown | 87 (64–111) | 91 (78–115) | 101 (76–126) |
| | gorget | 126 (93–151) | *149 (116–179)* | 102 (65–133) |
| | tail | 90 (67–112) | 89 (65–110) | 79 (50–102) |
| cortex thickness | crown | 133 (106–153) | 108 (95–126) | 146 (124–168) |
| | gorget | 173 (157–192) | *139 (111–164)* | 159 (129–199) |
| | tail | 64 (45–87) | 52 (30–78) | 52 (31–77) |
| keratin spacing | crown | 26 (15–36) | 32 (23–45) | 32 (24–44) |
| | gorget | 39 (25–66) | 39 (19–65) | 33 (20–47) |
| | tail | 25 (6–37) | *17 (12–22)* | 33 (19–54) |
| number of layers | crown | 13 (12–14) | *16 (12–21)* | 11 (7–14) |
| | gorget | 9 (9–10) | *12 (10–14)* | 8 (6–10) |
| | tail | 5 (4–6) | 3 (3–4) | 3 (2–3) |
| melanin thickness | crown | 37 (28–46) | *44 (34–53)* | 39 (32–46) |
| | gorget | 52 (41–65) | *49 (39–61)* | 52 (40–60) |
| | tail | 40 (32–48) | 44 (36–58) | 56 (43–68) |
| top mel. thickness | crown | 85 (67–107) | 91 (72–113) | 123 (100–152) |
| | gorget | 168 (141–190) | *198 (167–224)* | 174 (128–206) |
| | tail | 165 (132–207) | 165 (147–197) | 211 (186–229) |

for *H. gularis* (figure 1a) was predicted under both melanosome shape models (figure 4, electronic supplementary material, figure S5). For gorget feathers, secondary peaks at approximately 450 nm for both *H. gularis* and *H. branickii* (figure 1a,c) were predicted by both optical models (figure 4e, electronic supplementary material, figure S5E). Critically, this peak was correctly predicted as absent in the backcross hybrid spectra (figure 4e, arrow). The gorgets' modelled spectra were on average 119 nm red-shifted relative to the empirical spectra (compared with only 10 and 28 nm for crown and tail feathers, respectively; figure 4b,e). Differences in melanin refractive index among patches explained the greater discrepancy in gorget feather models relative to crown and tail feather models.

## 3.4. Genetic analyses suggest FMNH 511084 is a backcross hybrid

We collected $6.54 \times 10^5$ raw reads for the FMNH 511084 Illumina library. We retained 99.6% of the read data after trimming of low-quality bases and removal of adapter contamination. After quality control, we assembled cleaned reads from specimen FMNH 511084 into consensus contigs and identified 3763 UCE sequences with an average length of 1210 bp. Network analysis of UCE trees suggests FMNH 511084 shares a considerable amount of nuclear DNA with both *H. gularis* and *H. branickii* (figure 5a). Analysis of genome-wide SNP data revealed FMNH 511084 was assigned a 16.3% [95% CI: 15.2–17.5%] proportion of hybrid ancestry and an interspecific heterozygosity of 11% (figure 5b), suggestive of a late-generation hybrid or backcross (e.g. see [69]).

## 3.5. Comparative analysis documents rate of colour divergence in the backcross hybrid

Based on a recent phylogeny of all hummingbirds [36] and a colour dataset spanning hummingbirds [17], we estimated a rate of colour evolution ($\sigma^2$) of 0.0043 JNDs$^2$ My$^{-1}$. Using this rate estimate, we calculated the time needed to achieve the observed *H. gularis*-putative hybrid colour divergence as (0.207 JNDs)$^2$ / (0.0043 JNDs$^2$ My$^{-1}$). This calculation indicates it would take 10.0 My to achieve the observed difference in gorget coloration between *H. gularis* and the putative hybrid. A similar magnitude result was obtained for the divergence between *H. branickii* and the putative hybrid (6.6 My).

**Figure 3.** Colour and morphological divergence in *Heliodoxa* hummingbirds. Plots show perceptually uniform [82] avian tetrahedral colour space coordinates (*a–c*) and principal components (PC) axes 1 and 2 for feather morphological data (*d–f*) in crown (*a,d*), gorget (*b,e*) and tail feathers (*c,f*). Point colours correspond to *Heliodoxa branickii* (green), *Heliodoxa gularis* (purple) and the *Heliodoxa* backcross hybrid (orange).

## 3.6. Ecological niche modelling supports the potential for a hybrid zone in the Cordillera Azul

Ecological niche models indicated that there was little spatial area of overlap between these species when areas of high suitability are co-projected spatially (figure 2). Our niche tests fail to reject the null hypothesis of niche conservatism, with high support for nice similarity from the perspective of *H. gularis* ($p < 0.05$). These results are consistent with the hypothesis of niche conservatism between sister species in allopatry [70]. In our PC analyses, PC1 was most reliant on continentality and seasonality, whereas PC2 was most affected by elevation and annual precipitation (figure 2*b*). The backcross hybrid individual was found at a site that is close to the contact zone between suitable environments for *H. branickii* and *H. gularis*.

# 4. Discussion

We show that plumage coloration of FMNH 511084 is a novel phenotype (figure 1) that is morphologically (figure 3*e*) and perceptually (electronic supplementary material, figure S3) distinct from both *H. branickii* and *H. gularis*. Morphologically, we identify the nanostructural bases of variation in gorget coloration (e.g. thin cortex, broader surficial melanosomes), a prominent and divergent avian-perceivable trait between the hybrid and its parental taxa, namely the unique melanin arrangement, and predicted optimal models' outputs matching the observed reflectance spectra (figure 4*b,e*). Mitochondrial sequence data for FMNH 511084 are identical to a sample of *H. branickii* but distinct from *H. gularis.* This is in contrast to the similarity in mitogenomes reported for the genus *Coeligena* [71]. Nuclear data document differences between FMNH 511084, *H. branickii* and *H. gularis* (figure 5*a*), and increased heterozygosity in FMNH 511084 suggests it is a late-generation backcross

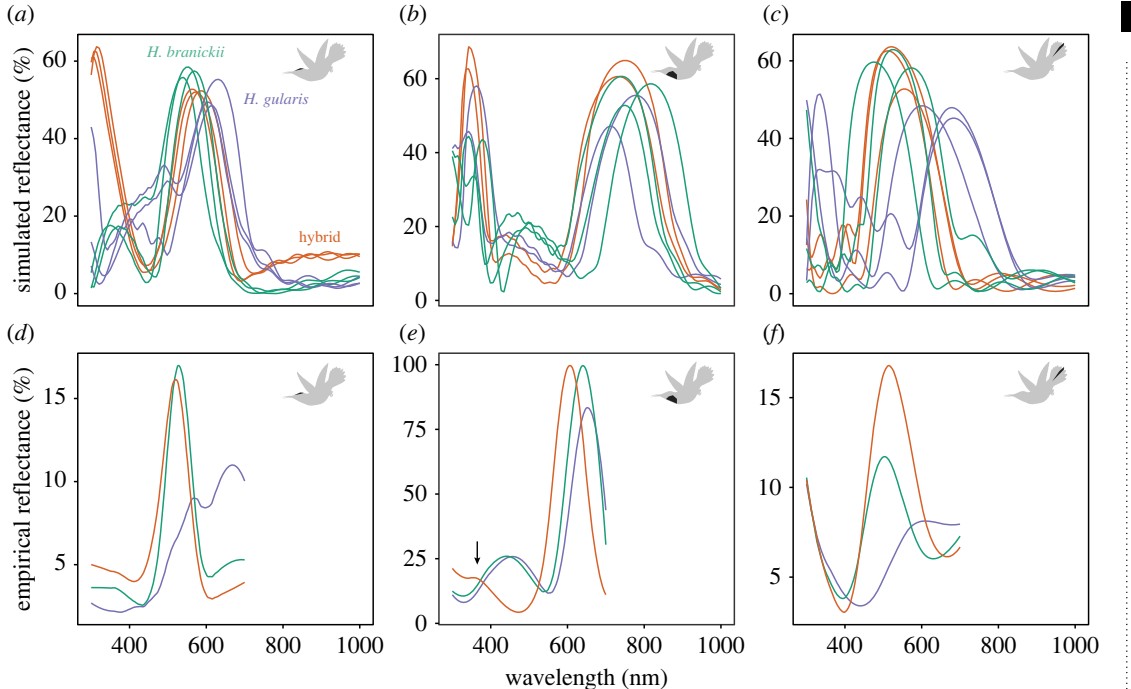

**Figure 4.** Optical modelling reveals how morphological divergence translates to colour divergence in hybridizing *Heliodoxa* species. Panels show simulated (*a–c*) and empirical (*d–f*) reflectance spectra for crown (*a,d*), gorget (*b,e*) and tail feathers (*c,f*). Upper panels show individual simulations for different barbules within a feather. Optical model assumes block-shaped air spaces within melanosomes. Colours correspond to *Heliodoxa branickii* (green), *Heliodoxa gularis* (purple) and the *Heliodoxa* backcross hybrid (orange).

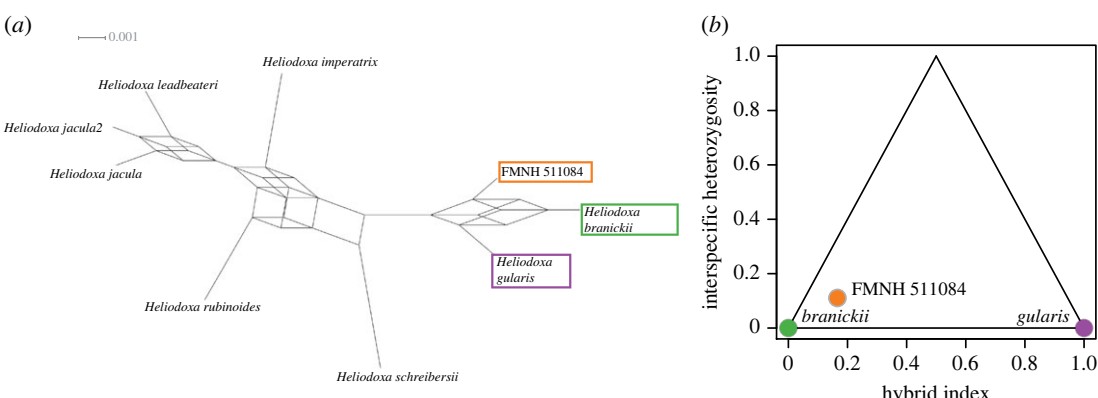

**Figure 5.** Hybrid ancestry in *Heliodoxa* hummingbirds. (*a*) Phylogenetic network plot produced using splitsTree (options: 0.1 edge cut-off, ConsensusNetwork) from a set of 3763 trees derived from ultra-conserved elements (UCEs) across the genus. Branch lengths are proportional to the weight of estimated splits among taxa (see legend). (*b*) Triangle plot shows the relationship between hybrid index and interspecific heterozygosity for 2031 single-nucleotide polymorphisms (SNPs). For interpretation: F1 individuals are expected to fall toward the top of the triangles, F2/late generation hybrids near the middle and backcrossed individuals near the sides. The location of FMNH 511084 suggests it is probably a backcross hybrid with *Heliodoxa branickii*.

with *H. branickii* (figure 5*b*). We suggest it is 'transgressive,' demonstrating that past hybridization has led to macrophenotypic shifts over contemporary timescales, outpacing non-hybridization-based character and ecological niche evolution by millions of years, especially in contact zones between related species that use nanostructural mechanisms to generate communicative coloration. This could be established with additional population-scale data. Individual FMNH 511084 is, to our knowledge, the first example in hummingbirds where intra-specific variation in plumage is not related to

geography. Other avian families characterized by iridescent plumages (e.g. Nectariidae, Paradisaeidae) also have no known examples of polymorphism in iridescent plumages.

The niche-modelling data for *H. branickii* and *H. gularis* predict that the two species are either locally sympatric or occasionally co-occur near the edges of their distributions in the Cordillera Azul (figure 2). The collection locality is within the known range of *H. branickii* and 200 km south of the known distribution of *H. gularis* [67]. The Río Huallaga runs from south to the north through the Andes before making an eastward turn to pass through the Andean foothills at the northern end of the Cordillera Azul and out into the Amazon Basin. The eastward stretch appears to act as an important biogeographic barrier, and there are no records to date of *H. branickii* north of the Huallaga nor of *H. gularis* south and east of the Huallaga. Even though there is no evidence for long-distance dispersal in these species, other sedentary (or elevational migrant) species of hummingbirds have exhibited an extraordinary propensity for vagrancy in more well-surveyed areas [72]. The lower elevational distributions of the two species differ [67], with *H. gularis* recorded to 250 m.a.s.l. and *H. branickii* to 650 m.a.s.l. While both species are poorly sampled throughout the Eastern Andes, this elevational difference may imply an increased ability of *H. gularis* to disperse across low-elevation river valleys which could include the Río Huallaga. Whereas our niche models indicate that areas of potential geographical overlap are rare, our ecological niche analyses suggest that areas of local parapatry between these species may exist in unsurveyed parts of the Cordillera Azul and the adjacent main slope of the adjacent Eastern Andes. Future sampling and observations will be required to determine more about the species' distributions in this region, and to determine whether resident populations of *H. gularis* exist south of the currently known distribution. Although we do not have genetic sequences from other *Heliodoxa* at the Pescadero site, another male specimen collected is a typical *H. branickii* morphologically.

One putative mechanism for the bright yellow gorget colour in the backcross hybrid is transgressive segregation, in which recombination occurs in genes with antagonistic effects. An example of this is the agouti-melanocortin pigment-based coloration system [6]. Structural colours are unique in that chemical properties of pigments do not primarily cause the observed colour, but rather the dimensions and arrangement of pigment granules, air bubbles and keratin layers define the colour [14]. The highly ordered stacks of melanosomes seen in iridescent bird feathers probably result from self-assembly [73]. Although we lack transcriptomic work needed to identify candidate genic or regulatory regions in the genome that can explain the developmental origins of iridescence, it is likely that upregulation of genes involved in keratin polymerization and melanosome shape (e.g. Pmel17) [74] may be critical in setting the stage for self-assembly to occur [73]. Interestingly, most nanostructural differences in FMNH 511084 are in the uppermost layer of feather barbules (e.g. thickness of the keratin cortex, diameter of surficial melanin platelets; table 2). Thus, another possible explanation for the transgressive colour of this individual is that the outer regions of feathers are more prone to environmental fluctuations during feather development, suggestive of genotype-by-environment effect on feather morphology and plumage coloration [6]. Future work combining functional genomics and materials science will be necessary to tease apart these scenarios in hummingbirds. In either case, our results are the first example of quantifying the effects of hybridization across scales—from feather nanostructure, to signal phenotype and ecological niche space.

Understanding the origins of phenotypic novelty remains an important question in evolutionary biology. Hummingbirds are textbook examples of diversity in acoustic [75], visual [16,17] and behavioural communication cues [76,77]. Precise coordination among these sensory modalities is probably key to effective mating displays [76]. On an evolutionary timescale, aspects of acoustic signals coevolve to drive diversity in the bee hummingbird clades [75]. Similarly, patterns of evolutionary coevolution in feather nanostructure traits seem to have partially driven the explosive diversity of visual signals across hummingbirds [17]. Our genetic data document that the hybrid is genetically distinct (figure 5) from the samples of the two species we currently have available to us. The Cordillera Azul where this individual was collected is an outlying foothill of the Andes where isolated populations could become genetically distinct (figure 2a). Geographical population structure may explain the nuclear distinctiveness of FMNH 511084. The genetic sample of *H. branickii* is from the slopes of the main Andes and could be a different population. One challenge with respect to assuming genetic tools will uncover hybridization is the potential that FMNH 511084 is not a recent hybrid, but part of a history of hybridization. However, this does not rule out that transgressive segregation has contributed to the diversity of signal phenotypes in this well-studied and charismatic clade of birds. Other bird lineages show extensive plumage colour variation despite being closely related (e.g. *Thalurania* hummingbirds, *Lepidothrix* manakins) [3,72,78]. Barrera-Guzman *et al.* [3]

studied the effects of hybridization on a single phenotype (crown coloration) in manakins (Aves: Pipridae). While FMNH 511084 is not an F1 hybrid, distinct gorget coloration made it recognizable from *H. branickii* and *H. gularis* (figure 1, electronic supplementary material, figures S2 and S3). If male gorget colour differences eventually spread through a population via female choice [79], this would constitute a rare example of 'Type I' hybrid speciation, whereby hybridization directly causes reproductive isolation [80]. This process differs from the golden-crowned manakin in which novel yellow crown colour evolved several generations after hybridization as new feather mutations accumulated [3]. However, unless mating preferences are also divergent, it is unlikely hybrid individuals will mate and produce viable offspring to facilitate this process. Nonetheless, our results highlight that analysis of colour in atypical individuals can provide insight into the mechanisms of how novel hybrid phenotypes are generated.

Ethics. Fieldwork was conducted through the Centro de Conservación, Investigación y Manejo de Areas Naturales (CIMA) and the Peruvian government following protocols approved by the Field Museum's animal care and use committee (protocol #12-2, to J.M.B.). Specimen FMNH 511084 was exported under SERFOR permit #18-1989, in accordance with all rules and regulations of the government of Peru and the U.D. government.

Data accessibility. Genomic data (UCE alignments, VCF file with SNPs), SEM images, trait datasets and R code needed to reproduce analyses have all been uploaded to Dryad (https://doi.org/10.5061/dryad.pk0p2ngs7) [42].

The data are provided in electronic supplementary material [83].

Authors' contributions. C.M.E.: conceptualization, data curation, formal analysis, investigation, methodology, project administration, software, supervision, visualization, writing—original draft, writing—review and editing; J.C.C.: data curation, formal analysis, visualization, writing—original draft; S.J.H.: data curation, writing—original draft; E.Z.: data curation; T.Z.P.S.: investigation; J.D.M.: data curation, writing—review and editing; T.H.: data curation, formal analysis; M.E.H.: funding acquisition, supervision, writing—original draft, writing—review and editing; J.M.B.: conceptualization, data curation, resources, supervision, writing—original draft, writing—review and editing.

Conflict of interest declaration. The authors declare no conflict of interest.

Funding. This work was partially supported by a Grainger Bioinformatics Postdoctoral Fellowship (to C.M.E.), the National Science Foundation (grant no. NSF EP-2112468, to C.M.E. and S.J.H.), and by the University of Kansas Institutional Research and Career Advancement award (J.C.C., NIH #5K12GM063651-20). Additional funding was received by the Harley Jones Van Cleave Prof. (to M.E.H.). We are grateful to Rauri Bowie, Jimmy McGuire and Ammon Corl for providing access to genomic data collected as part of a larger funded project (NSF EP-1831833). Fieldwork was funded by the H. B. Conover fund (Field Museum), the Emerging Pathogens Project (to S.J.H.) and NSF DEB-1120054 (to J.M.B.).

Acknowledgements. We thank Scott Robinson and Catherine Wallace (University of Illinois Urbana-Champaign) for assistance with electron microscopy. We thank Jorge Soberón and A. Townsend Peterson for providing information and code related to ecological niche modelling. We thank Patricia Alverez Loayza, Natalia Piland, Luis Cueto, Heather Skeen, Karen Verde, Shane DuBay, Willy Nuñez and CORBIDI for assistance with logistics in Peru and with facilitating exportation permits.

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
