## [Peer Review File · Royal Society Open Science]

Review History

RSOS-210766.R0 (Original submission)

Review form: Reviewer 1 (Bodo Wilts)

Is the manuscript scientifically sound in its present form?

Yes

Are the interpretations and conclusions justified by the results?

No

Is the language acceptable?

Yes

Do you have any ethical concerns with this paper?

No

Have you any concerns about statistical analyses in this paper?

No

Recommendation?

Accept with minor revision (please list in comments)

Comments to the Author(s)

Eliason and colleagues in their paper "Transgressive hybridization causes rapid gorget color divergence in *Heliodoxa* hummingbirds (Aves: Trochilidae)" present a nanostructural and ecological analysis of a hybrid hummingbird found in Peru. A combination of optical modelling with electron microscopical and reflectance analysis shows that the hybrid bird has a different coloration than their parents and the authors well describe the reasons and effects of hybridisation on understanding bird coloration. The manuscript is a pleasure to read and I think well suited for RSOS.

I only found a few things that the authors should consider in a revised version.

The nanostructural and spectral data is nice and clear, unfortunately, figure 2 is not at all discussed in the text as well as details on the spectra. I would ask the authors to add more information on the spectral properties and the analysis, so that readers can follow this section. For me, it is really condensed and hard to follow. I would also suggest to add the key essence of SI Table S2 to the main text, as this is quite instrumental to the results and the resulting discussion.

I also suggest to add broader references on structural coloration, particularly on structural colours (work by Gomez on hummingbirds, or work by Tinbergen on parrot colours for colour mixing, etc.).

The modelling work (lines 287-304) is very descriptive, which is OK for the target audience, but somewhat irritating when stating in line 313 "we identify the nanostructural basis of the gorget". Maybe a simple rephrase would do, but I'd like to see a more detailed analysis in the results, if possible.

Figure 4 prints really badly without axes etc. Please add axes on all panels.

Minor things:

L. 91: There is a verb missing before ecological? Sentence does not read well.

L. 255: Add reference to Figure 1B.

L. 276 pp: Figure 2 is not discussed and I think this section should be considerably extended in the light of this.

L. 350: Please add references to support this statement.

L. 358: Another point why table S2 should be in the main text.

All in all, this is a nice paper and I hope that these points help making it even more appealing.
Bodo Wilts

Review form: Reviewer 2

Is the manuscript scientifically sound in its present form?

No

Are the interpretations and conclusions justified by the results?

No

Is the language acceptable?

Yes

Do you have any ethical concerns with this paper?

No

Have you any concerns about statistical analyses in this paper?

No

Recommendation?

Reject

Comments to the Author(s)

This ms describes a hybrid hummingbird whose iridescent plumage color especially on its gorget (upper throat) lies outside the range of the parental species. Such transgressive phenotypes are relevant to understanding the role of hybridisation and interspecific gene flow in evolution. The main focus of the study is on describing the plumage color and showing that it can be attributed to the underlying nanostructure of the feathers (Figs 2-4). This work is done well and there are some observations about the specific location of the structural changes and speculation that there might be a '*genotype-by-environment effect on feather morphology*'. However as the genetics of iridescent coloration is little known - in contrast to some types of pigment coloration - the work remains primarily a routine description of a structural color. Although it makes an interesting observation in describing the hybrid, and highlights open questions about genetics of structural coloration the ms lacks the broad significance and definite conclusions expected of a general interest journal.

Minor comments.

1. Fig. 2. Explain how the colours were recorded. Is the angular separation of the light source was fixed relative to the line of sight of the viewer, as this angle can have a major effect on the spectral peak (hue) of reflected light?
2. Line 267. What does the estimate of the rate of color evolution in hummingbirds show? Presumably this depends substantially on the strength of selection?
2. Line 348. Should read 'Structural colors are unique!' Not all such color are iridescent.

Decision letter (RSOS-210766.R0)

Dear Dr Eliason

The Editors assigned to your paper RSOS-210766 "Transgressive hybridization causes rapid gorget color divergence in *Heliodoxa* hummingbirds (Aves: Trochilidae)" have made a decision based on their reading of the paper and any comments received from reviewers.

Regrettably, in view of the reports received, the manuscript has been rejected in its current form. However, a new manuscript may be submitted which takes into consideration these comments.

We invite you to respond to the comments supplied below and prepare a resubmission of your manuscript. Below the referees' and Editors' comments (where applicable) we provide additional requirements. We provide guidance below to help you prepare your revision.

Please note that resubmitting your manuscript does not guarantee eventual acceptance, and we do not generally allow multiple rounds of revision and resubmission, so we urge you to make every effort to fully address all of the comments at this stage. If deemed necessary by the Editors, your manuscript will be sent back to one or more of the original reviewers for assessment. If the original reviewers are not available, we may invite new reviewers.

Please resubmit your revised manuscript and required files (see below) no later than 25-Nov-2021. Note: the ScholarOne system will 'lock' if resubmission is attempted on or after this deadline. If you do not think you will be able to meet this deadline, please contact the editorial office immediately.

Please note article processing charges apply to papers accepted for publication in Royal Society Open Science (<https://royalsocietypublishing.org/rsos/charges>). Charges will also apply to papers transferred to the journal from other Royal Society Publishing journals, as well as papers submitted as part of our collaboration with the Royal Society of Chemistry (<https://royalsocietypublishing.org/rsos/chemistry>). Fee waivers are available but must be requested when you submit your manuscript (<https://royalsocietypublishing.org/rsos/waivers>).

Thank you for submitting your manuscript to Royal Society Open Science and we look forward to receiving your resubmission. If you have any questions at all, please do not hesitate to get in touch.

on behalf of Dr Kristina Sefc (Associate Editor) and Kevin Padian (Subject Editor)
openscience@royalsociety.org

Subject Editor Comments to Author (Professor Kevin Padian):
Comments to the Author:

Thanks for your submission. As you will see there is general support for your work but a variety of concerns. (The view that it is too narrow is not a concern for RSOS.) One reviewer and also our AE had questions that appear to require addressing before we can consider the work further. However we encourage you to address these and resubmit.

Also there was a concern raised in the review process about whether the collection and export of the specimen was in accordance with international regulations, which of course have varied through time, among nations, and with specific taxa. Could you please just include for us some information on the collection and export of the specimen? Thanks and I look forward to a resubmission.

Associate Editor Comments to Author (Dr Kristina Sefc):
Comments to the Author:

We have responses from two reviewers, which provide a somewhat mixed assessment of the manuscript. Reviewer 1 requests more information on the spectral analysis of the coloration, but is overall positive about the suitability of the manuscript for RSOS. Reviewer 2 is concerned that the limited scope of the study makes it unsuitable for RSOS, but has no major criticism regarding the technical approach and data interpretation. Given that so little is currently known about structural coloration, I feel that the manuscript could serve as an anchor point for future studies, and therefore recommend that it be considered for RSOS. I do, however, have one concern

regarding a central claim of the manuscript. The manuscript relies on the assumption that the specimen is an F1 hybrid between the suggested parental species. Based on what is presented in the manuscript, this claim is not yet convincing, and more information about the genetic data used to confirm the hybrid origin is needed. First, regarding the supposed hybridization event (irrespective of the hybrid generation represented by the specimen), please report how you assessed intraspecific variability of the nuclear marker (in presumed parental species) to make sure that the detected polymorphisms are diagnostic for *H. gularis*. The reference to the manakin paper (ref. #8) in line 247 seems arbitrary, please explain how it contributes in this context or remove it.

Then, regarding the proposed F1 hybrid generation represented by the specimen, I'm not sure that I see any evidence for this in the manuscript. The genetic data (if species diagnostic) would also support less recent introgression. An estimate of the hybrid generation represented by the specimen could be achieved with multilocus nuclear data.

Also, please report the length of the sequenced DNA fragments, and explain in line 248 whether the divergence estimate is based on your genetic data, or including the data from the cited reference?

I would like to ask the authors to provide a firm backup for the proposed hybrid origin and generation (if necessary, by producing more genetic data) and to follow the suggestions given by the two referees in a revision of their manuscript.

Reviewer comments to Author:

Reviewer: 1

Comments to the Author(s)

Eliason and colleagues in their paper "Transgressive hybridization causes rapid gorget color divergence in *Heliodoxa* hummingbirds (Aves: Trochilidae)" present a nanostructural and ecological analysis of a hybrid hummingbird found in Peru. A combination of optical modelling with electron microscopical and reflectance analysis shows that the hybrid bird has a different coloration than their parents and the authors well describe the reasons and effects of hybridisation on understanding bird coloration. The manuscript is a pleasure to read and I think well suited for RSOS.

I only found a few things that the authors should consider in a revised version.

The nanostructural and spectral data is nice and clear, unfortunately, figure 2 is not at all discussed in the text as well as details on the spectra. I would ask the authors to add more information on the spectral properties and the analysis, so that readers can follow this section. For me, it is really condensed and hard to follow. I would also suggest to add the key essence of SI Table S2 to the main text, as this is quite instrumental to the results and the resulting discussion.

I also suggest to add broader references on structural coloration, particularly on structural colours (work by Gomez on hummingbirds, or work by Tinbergen on parrot colours for colour mixing, etc.).

The modelling work (lines 287-304) is very descriptive, which is OK for the target audience, but somewhat irritating when stating in line 313 "we identify the nanostructural basis of the gorget". Maybe a simple rephrase would do, but I'd like to see a more detailed analysis in the results, if possible.

Figure 4 prints really badly without axes etc. Please add axes on all panels.

Minor things:

L. 91: There is a verb missing before ecological? Sentence does not read well.

L. 255: Add reference to Figure 1B.

L. 276 pp: Figure 2 is not discussed and I think this section should be considerably extended in the light of this.

L. 350: Please add references to support this statement.

L. 358: Another point why table S2 should be in the main text.

All in all, this is a nice paper and I hope that these points help making it even more appealing.
Bodo Wilts

Reviewer: 2

Comments to the Author(s)

This ms describes a hybrid hummingbird whose iridescent plumage color especially on its gorget (upper throat) lies outside the range of the parental species. Such transgressive phenotypes are relevant to understanding the role of hybridisation and interspecific gene flow in evolution. The main focus of the study is on describing the plumage color and showing that it can be attributed to the underlying nanostructure of the feathers (Figs 2-4). This work is done well and there are some observations about the specific location of the structural changes and speculation that there might be a '*genotype-by-environment effect on feather morphology*'. However as the genetics of iridescent coloration is little known - in contrast to some types of pigment coloration - the work remains primarily a routine description of a structural color. Although it makes an interesting observation in describing the hybrid, and highlights open questions about genetics of structural coloration the ms lacks the broad significance and definite conclusions expected of a general interest journal.

Minor comments.

1. Fig. 2. Explain how the colours were recorded. Is the angular separation of the light source was fixed relative to the line of sight of the viewer, as this angle can have a major effect on the spectral peak (hue) of reflected light?
2. Line 267. What does the estimate of the rate of color evolution in hummingbirds show? Presumably this depends substantially on the strength of selection?
2. Line 348. Should read 'Structural colors are unique!' Not all such color are iridescent.

===PREPARING YOUR MANUSCRIPT===

===PREPARING YOUR REVISION IN SCHOLARONE===

Author's Response to Decision Letter for (RSOS-210766.R0)

See Appendix A.

Decision letter (RSOS-221603.R0)

Dear Dr Eliason

On behalf of the Editors, we are pleased to inform you that your Manuscript RSOS-221603 "Interspecific hybridization explains rapid gorget color divergence in *Heliodoxa* hummingbirds (Aves: Trochilidae)" has been accepted for publication in Royal Society Open Science subject to minor revision. The specific revisions are outlined immediately below:

1. You indicate that no ethical approvals were sought/required for this work; however, we note that you conducted fieldwork and sampling on an avian population. These would generally require approval to be conducted. Please can you provide evidence that permissions to conduct the fieldwork and ethical approvals to conduct the sampling work?

If no permissions were sought or required, please can you explicitly state why this was the case in your ethical approvals statement in the manuscript? (If none were sought or required, it would be helpful if you can supply some form of evidence to support this.)

2. You indicate that your datasets will be uploaded to appropriate repositories subject to acceptance, please can we ask that you now do this?

Please submit your revised manuscript and required files (see below) no later than 7 days from today's (ie 17-Jan-2023) date. Note: the ScholarOne system will 'lock' if submission of the revision is attempted 7 or more days after the deadline. If you do not think you will be able to meet this deadline please contact the editorial office immediately.

Please note article processing charges apply to papers accepted for publication in Royal Society Open Science (<https://royalsocietypublishing.org/rsos/charges>). Charges will also apply to papers transferred to the journal from other Royal Society Publishing journals, as well as papers submitted as part of our collaboration with the Royal Society of Chemistry

(<https://royalsocietypublishing.org/rsos/chemistry>). Fee waivers are available but must be requested when you submit your revision (<https://royalsocietypublishing.org/rsos/waivers>).

on behalf of Dr Kristina Sefc (Associate Editor) and Kevin Padian (Subject Editor)
openscience@royalsociety.org

Associate Editor Comments to Author (Dr Kristina Sefc):

The revision addresses all of the previous comments very carefully, and I am happy to recommend this nice manuscript for publication.

===PREPARING YOUR MANUSCRIPT===

one version should clearly identify all the changes that have been made (for instance, in coloured highlight, in bold text, or tracked changes);

Please ensure that you include an acknowledgements' section before your reference list/bibliography. This should acknowledge anyone who assisted with your work, but does not qualify as an author per the guidelines at <https://royalsociety.org/journals/ethics-policies/openness/.s>

===PREPARING YOUR REVISION IN SCHOLARONE===

-- If you are requesting an article processing charge waiver, you must select the relevant waiver option (if requesting a discretionary waiver, the form should have been uploaded, see 'File upload' above).

-- If you have uploaded any electronic supplementary (ESM) files, please ensure you follow the guidance at <https://royalsociety.org/journals/authors/author-guidelines/#supplementary-material> to include a suitable title and informative caption. An example of appropriate titling and captioning may be found at https://figshare.com/articles/Table_S2_from_Is_there_a_trade-off_between_peak_performance_and_performance_breadth_across_temperatures_for_aerobic_scope_in_teleost_fishes_/3843624.

Author's Response to Decision Letter for (RSOS-221603.R0)

See Appendix B.

Decision letter (RSOS-221603.R1)

Dear Dr Eliason:

I am pleased to inform you that your manuscript entitled "Interspecific hybridization explains rapid gorget color divergence in *Heliodoxa* hummingbirds (Aves: Trochilidae)" is now accepted for publication in Royal Society Open Science.

Please remember to make any data sets or code libraries 'live' prior to publication, and update any links as needed when you receive a proof to check - for instance, from a private 'for review' URL to a publicly accessible 'for publication' URL. It is also good practice to add data sets, code and other digital materials to your reference list.

Royal Society Open Science is a fully open access journal. A payment may be due before your article is published. The Royal Society has partnered with Copyright Clearance Center's (CCC's) RightsLink service to allow authors to pay article processing charges or page charges. After your manuscript has been accepted, the corresponding author will receive an email from CCC with the subject "Please submit your article processing/open access charge(s)/page charges" inviting you to pay your charges or request an invoice. The email from CCC will come from the email domain @copyright.com (if you have any queries regarding fees, please see <https://royalsocietypublishing.org/rsos/charges> or contact authorfees@royalsociety.org). If you request an invoice, it will be sent to you from CCC. It is important to be cautious about payment scams. If you receive an email or text message requesting payment and have any concerns, we recommend contacting us through our website, rather than clicking on any links. **The Society will never ask you to make a direct payment.**

on behalf of Dr Kristina Sefc (Associate Editor) and Professor Kevin Padian (Subject Editor).

Associate Editor Dr Kristina Sefc Comments to Author:

Associate Editor: 1
Comments to the Author:
(There are no comments.)

Reviewer(s)' Comments to Author:
Follow Royal Society Publishing on Twitter: @RSocPublishing
Follow Royal Society Publishing on Facebook:
<https://www.facebook.com/RoyalSocietyPublishing/>
Read Royal Society Publishing's blog:
<https://royalsociety.org/blog/blogsearchpage/?category=Publishing>

Appendix A

Dear Editorial Board Members of RSOS:

We are grateful for the constructive evaluation of our work and have made the following changes, noted in bold, to the manuscript throughout its text. We hope that the revised version is suitable for publication in RSOS.

Sincerely, Chad Eliason and co-authors.

--

Subject Editor Comments to Author (Professor Kevin Padian):

Comments to the Author:

Thanks for your submission. As you will see there is general support for your work but a variety of concerns. (The view that it is too narrow is not a concern for RSOS.) One reviewer and also our AE had questions that appear to require addressing before we can consider the work further. However we encourage you to address these and resubmit.

Also there was a concern raised in the review process about whether the collection and export of the specimen was in accordance with international regulations, which of course have varied through time, among nations, and with specific taxa. Could you please just include for us some information on the collection and export of the specimen? Thanks and I look forward to a resubmission.

Our Response: We originally stated the permit number in the acknowledgements, but we have now added a line that we followed all rules and regulations (L460-462: "Specimen FMNH 511084 was exported under SERFOR permit #18-1989. These samples were exported following all rules and regulations of the government of Peru and the U.D. government.").

--

Associate Editor Comments to Author (Dr Kristina Sefc):

Comments to the Author:

We have responses from two reviewers, which provide a somewhat mixed assessment of the manuscript. Reviewer 1 requests more information on the spectral analysis of the coloration, but is overall positive about the suitability of the manuscript for RSOS. Reviewer 2 is concerned that the limited scope of the study makes it unsuitable for RSOS, but has no major criticism regarding the technical approach and data interpretation. Given that so little is currently known about structural coloration, I feel that the manuscript could serve as an anchor point for future studies, and therefore recommend that it be considered for RSOS.

Our Response: We greatly appreciate this enthusiasm about our work.

I do, however, have one concern regarding a central claim of the manuscript. The manuscript relies on the assumption that the specimen is an F1 hybrid between the suggested parental species. Based on what is presented in the manuscript, this claim is not yet convincing, and more information about the genetic data used to confirm the hybrid origin is needed. First, regarding the supposed hybridization event (irrespective of the hybrid generation represented by the specimen), please report how you assessed intraspecific variability of the nuclear marker (in presumed parental species) to make sure that the detected polymorphisms are diagnostic for *H. gularis*.

Our Response: We have now added new genomic data for the putative hybrid individual and a multilocus data set for *Heliodoxa* hummingbirds. Based on a splitsTree analysis and assessment of hybrid index/heterozygosity in the putative hybrid (new Fig. 5), we have revised the text considerably to reflect the result that the individual is likely of hybrid origin (late-generation, or backcross with *H. branickii*) rather than an F1 (e.g., L41, 302, 315, 324). Nonetheless, even under this new interpretation of the origin of the individual, we still feel that our primary results showing (1) the nanostructural basis for structural coloration in closely-related individuals and (2) rapid color change in contemporary timescales (several generations) are well-supported and represent a considerable advance in the study of animal coloration.

The reference to the manakin paper (ref. #8) in line 247 seems arbitrary, please explain how it contributes in this context or remove it.

Our Response: Done. We removed the citation.

Then, regarding the proposed F1 hybrid generation represented by the specimen, I'm not sure that I see any evidence for this in the manuscript. The genetic data (if species diagnostic) would also support less recent introgression. An estimate of the hybrid generation represented by the specimen could be achieved with multilocus nuclear data.

Our Response: We appreciate this point, and have now added new data (whole genome for the individual, multilocus UCE data set for *Heliodoxa*; see Methods L207-231) and results (hybrid index, phylogenetic network; L315-324). These new analyses suggest that the individual is NOT an F1 hybrid, rather it is the product of less recent backcross/introgression with the two parental species.

Also, please report the length of the sequenced DNA fragments, and explain in line 248 whether the divergence estimate is based on your genetic data, or including the data from the cited reference?

Our Response: This line has been removed in the new version of the manuscript.

I would like to ask the authors to provide a firm backup for the proposed hybrid origin and generation (if necessary, by producing more genetic data) and to follow the suggestions given by the two referees in a revision of their manuscript.

Our Response: We appreciate the comments, and we have followed your suggestion along with those of the two other referees (see below).

--

Reviewer comments to Author:

Reviewer: 1

Comments to the Author(s)

Eliason and colleagues in their paper "Transgressive hybridization causes rapid gorget color divergence in *Heliodoxa* hummingbirds (Aves: Trochilidae)" present a nanostructural and ecological analysis of a hybrid hummingbird found in Peru. A combination of optical modelling with electron microscopical and reflectance analysis shows that the hybrid bird has a different coloration than their parents and the authors well describe the reasons and effects of hybridisation on understanding bird coloration. The manuscript is a pleasure to read and I think well suited for RSOS.

Our Response: We greatly appreciate this positive comment about the readability of our manuscript.

I only found a few things that the authors should consider in a revised version.

The nanostructural and spectral data is nice and clear, unfortunately, figure 2 is not at all discussed in the text as well as details on the spectra. I would ask the authors to add more information on the spectral properties and the analysis, so that readers can follow this section. For me, it is really condensed and hard to follow.

Our Response: We have added more detail about our spectrophotometry setup (e.g., L136-141: "we measured reflectance with both the light source and spectrophotometer probe at a 90° angle with respect to the feather (i.e., normal incidence), as well as at the angle for which we observed maximal reflectance, which was variable among specimens. The latter metric has been shown to be more reliable, especially for iridescent plumages [23], therefore we used spectra recorded at the optimal incidence angle for downstream analyses.").

We also discuss Figure 1 (old Figure 2) in several places (e.g., LL 129, 305, 349, 440).

I would also suggest to add the key essence of SI Table S2 to the main text, as this is quite instrumental to the results and the resulting discussion.

Our Response: We moved Table S2 to the main text as Table 2.

I also suggest to add broader references on structural coloration, particularly on structural colours (work by Gomez on hummingbirds, or work by Tinbergen on parrot colours for colour mixing, etc.).

Our Response: We have added additional citations to Prum (2006)'s Bird Coloration chapter and Tinbergen and Wilts (2013) on spectral tuning in parrots (e.g., L72).

The modelling work (lines 287-304) is very descriptive, which is OK for the target audience, but somewhat irritating when stating in line 313 “we identify the nanostructural basis of the gorget”. Maybe a simple rephrase would do, but I’d like to see a more detailed analysis in the results, if possible.

Our Response: We elaborated on this point in L351: "we identify the nanostructural bases of variation in gorget coloration (e.g., thin cortex, broader surficial melanosomes)"

Figure 4 prints really badly without axes etc. Please add axes on all panels.

Done.

Minor things:

L. 91: There is a verb missing before ecological? Sentence does not read well.

Our Response: We agree that this did not read well. We changed it to "We also present ecological niche modeling to better understand a potentially important contact zone between *Heliodoxa taxa* in Peru" in L105-107.

L. 255: Add reference to Figure 1B.

Our Response: We have added it (now a reference to Fig. 2b) in L343.

L. 276 pp: Figure 2 is not discussed and I think this section should be considerably extended in the light of this.

Our Response: We discuss this figure (now Figure 1) 5 times. We also added several references to new Table 2 (L288, 290, 292-93, 409) as well as expanded the subheading ("Morphological divergence in key color-producing traits") and added details about the repeatability of these morphological measurements (L394).

L. 350: Please add references to support this statement.

Our Response: Done. We added a citation to a book chapter (Prum 2006) that surveys structural color mechanisms in bird feathers.

L. 358: Another point why table S2 should be in the main text.

Our Response: We moved Table S2 to the main text (new Table 2) and updated references in the text to match.

All in all, this is a nice paper and I hope that these points help making it even more appealing.

Our Response: Thank you for the positive feedback!

Bodo Wilts

--

Reviewer: 2

Comments to the Author(s)

This ms describes a hybrid hummingbird whose iridescent plumage color especially on its gorget (upper throat) lies outside the range of the parental species. Such transgressive phenotypes are relevant to understanding the role of hybridisation and interspecific gene flow in evolution. The main focus of the study is on describing the plumage color and showing that it can be attributed to the underlying nanostructure of the feathers (Figs 2-4). This work is done well and there are some observations about the specific location of the structural changes and speculation that there might be a "*genotype-by-environment effect on feather morphology*". However as the genetics of iridescent coloration is little known - in contrast to some types of pigment coloration - the work remains primarily a routine description of a structural color. Although it makes an interesting observation in describing the hybrid, and highlights open questions about genetics of structural coloration the ms lacks the broad significance and definite conclusions expected of a general interest journal.

Our Response: While we appreciate that we have not uncovered the genomic basis of the hybrid throat color, we politely disagree that this is a "routine description" given that we take the next steps in comparing the divergence to evolutionary rates at the clade level and incorporate detailed analyses/knowledge of the niches of these birds.

Minor comments.

1. Fig. 2. Explain how the colours were recorded. Is the angular separation of the light source was fixed relative to the line of sight of the viewer, as this angle can have a major effect on the spectral peak (hue) of reflected light?

Our Response: Yes, the angular separation was fixed as we used a bifurcated fiberoptic cables. We determined the angle feather tile to achieve maximum reflectance following Meadows et al. (2011).

We have now added these important details to the Methods, along with the citation to Meadows et al. (2011) near L136: "we measured reflectance using a bifurcated fiberoptic probe with both the light source and spectrophotometer probe at a 90° angle with respect to the feather (i.e., normal incidence), as well as at the angle for which we observed maximal reflectance, which was variable among specimens. The latter measurement geometry has been shown to be more reliable, especially for iridescent plumages [23], therefore we used spectra recorded at the optimal incidence angle for downstream analyses."

2. Line 267. What does the estimate of the rate of color evolution in hummingbirds show? Presumably this depends substantially on the strength of selection?

Our Response: We opted to use a Brownian motion model because i) it has fewer parameters/assumptions and ii) it is used in the approach we cite of calculating multivariate rates at each node using independent contrasts (e.g., McPeck et al. 2008 ref. 39). However, we agree that strong stabilizing selection can certainly influence the evolution of trait variation through time, as often inferred using Ornstein-Uhlenbeck (OU) models. Therefore, we have added an additional analysis that demonstrates significant phylogenetic signal in the color data (L196-197: "Given the significant level of phylogenetic signal in the color data ($K = 0.53$, $P < 0.01$) estimated with physignal [37]") to support our use of a Brownian motion model.

2. Line 348. Should read "Structural colors are unique" Not all such color are iridescent.

Our Response: We appreciate the referee pointing this out. We have changed the text as suggested.

Appendix B

Dear Dr Eliason

On behalf of the Editors, we are pleased to inform you that your Manuscript RSOS-221603 "Interspecific hybridization explains rapid gorget color divergence in *Heliodoxa* hummingbirds (Aves: Trochilidae)" has been accepted for publication in Royal Society Open Science subject to minor revision. The specific revisions are outlined immediately below:

1. You indicate that no ethical approvals were sought/required for this work; however, we note that you conducted fieldwork and sampling on an avian population. These would generally require approval to be conducted. Please can you provide evidence that permissions to conduct the fieldwork and ethical approvals to conduct the sampling work?

If no permissions were sought or required, please can you explicitly state why this was the case in your ethical approvals statement in the manuscript? (If none were sought or required, it would be helpful if you can supply some form of evidence to support this.)

Our response: We have added additional details about the approved protocol under which the field work was conducted (e.g., L461-64: "Field work was conducted...following protocols approved by the Field Museum's Animal Care and use committee (protocol #12-2, to JMB).").

2. You indicate that your datasets will be uploaded to appropriate repositories subject to acceptance, please can we ask that you now do this?

Our response: We have uploaded the data and code to replicate our analyses to Dryad. We also added a "Data Availability" statement at the end of the manuscript that provided the DOI link.

Please submit your revised manuscript and required files (see below) no later than 7 days from today's (ie 17-Jan-2023) date. Note: the ScholarOne system will 'lock' if submission of the revision is attempted 7 or more days after the deadline. If you do not think you will be able to meet this deadline please contact the editorial office immediately.

Thank you for submitting your manuscript to Royal Society Open Science and we look forward to receiving your revision. If you have any questions at all, please do not

hesitate to get in touch.

on behalf of Dr Kristina Sefc (Associate Editor) and Kevin Padian (Subject Editor)
openscience@royalsociety.org

Associate Editor Comments to Author (Dr Kristina Sefc):

The revision addresses all of the previous comments very carefully, and I am happy to recommend this nice manuscript for publication.